# Medical abortion drug dispensing practices among private pharmacy workers in Nepal: A mystery client study

**Anil Sigdel, Mirak Raj Angdembe, Pratik Khanal\*, Nilaramba Adhikari, Alina Maharjan, Mahesh Paudel**

Population Services International, Lalitpur, Nepal

\* iampratikkhanal@gmail.com

**Data Availability Statement:** All relevant data are within the manuscript.

**Funding:** The study is funded by the Women's Health project implemented by Population Services

## Abstract

### Background

Pharmacies are the first point of contact for women seeking medical abortion (MA) and act as important sources of information and referral in Nepal. Over the counter sale of MA drugs is not currently allowed in Nepal. This study aimed to assess the MA drug dispensing practices of pharmacy workers using mystery clients in Nepal.

### Methods

A cross-sectional study using the mystery client approach was conducted in 266 pharmacies in September-October 2019. These pharmacies had either received harm reduction training or medical detailing visits. A total of 532 visits were conducted by six male and six female mystery clients. Mystery clients without prescription approached the sample pharmacy and filled out a standard digital survey questionnaire using the SurveyCTO application immediately after each interaction.

### Results

Pharmacy workers dispensed MA drugs in 35.7% of the visits while they refused to provide MA drugs to the mystery clients in 39.3% of visits. Lack of evidence of prior consultation with a physician (27.5%), referral to other health facilities (21.8%), unavailability of MA drugs in the pharmacy (21.3%) and lack of prescription (16.4%) were the main reasons for refusal. Seventy percent of the pharmacy workers inquired clients about last menstrual period/months of pregnancy while 38.1% asked whether the pregnancy status was confirmed. During 65.1% of the visits, mystery clients were told about when to take the MA drugs while in 66.4% of visits, they were told about the route of drug administration. Similarly, mystery clients were briefed about what to expect during the abortion process in half of the visits, and information about the possible side effects of the MA drug was provided in 55.9% of the visits. Pharmacy workers provided correct information on taking MA drugs to mystery clients in 70.7% of visits.

International (PSI)/Nepal office. Neither the study nor the author's salaries are funded in whole or in part by a tobacco company. All the authors are PSI Nepal staffs. The authors declare no competing interests in relation to this work. The donors are anonymous and are related to philanthropic organizations. The authors are not aware of any competing interests.

**Competing interests:** The authors have declared that no competing interests exist.

## Conclusion

Despite legal provision of sale of MA drugs only on prescription, pharmacy workers dispensed MA drug in one out of three visits. As pharmacies are the initial contacts of women for abortion services in Nepal, correct supplementary information through pharmacy workers can be an effective strategy to expand access to quality safe abortion services.

## Introduction

There were around 121 million unintended pregnancies each year globally between 2015 and 2019 [1]. Of these unintended pregnancies, about 61% ended in abortion, which translates to 73 million abortions per year [1]. Among these abortions, 45% were unsafe and occurred globally each year, mostly (97%) in developing countries in Africa, Asia, and Latin America [2]. Unsafe abortion was one of the leading causes of maternal mortality and morbidity globally, accounting for about 18% of maternal deaths in the developing world in 2012 [3, 4].

The government of Nepal legalized abortion in 2002 [5] and medical abortion (MA) in 2009 [6]. The guideline to allow MA drugs through mid-level providers (paramedics) was introduced in the same year [6]. Under the current law, abortion is legalized up to 12 weeks gestational age on the request of pregnant women and up to 28 weeks of gestational age in the case of rape or incest, if the pregnancy is detrimental to a woman's health and life or if there is fetal impairment [7, 8]. Of all unintended pregnancies in Nepal, 69% end in abortion [9]. Out of an estimated 323,100 abortions performed in Nepal during 2014, about 58% (186,100) were considered unsafe i.e., conducted by untrained or unregistered providers or self-induced [10]. Unsafe abortion accounted for about 7% of maternal mortality in 2009, with most deaths in rural regions in Nepal [11].

In Nepal, government registered MA drug brands (combination regimen) are available only on prescription through government-accredited safe abortion providers [12, 13]. The new safe abortion services guideline of the Ministry of Health and Population (MoHP) in 2021 allowed the provision of MA by skilled birth attendants (SBAs) and trained auxiliary nurse midwives (ANMs) up to ten weeks of gestation [8]. Combination MA drugs contain mifepristone and misoprostol, the two drugs when used one after the other can cause termination of pregnancy [14]. According to the WHO [15], the combination regimen of MA drugs (mifepristone and misoprostol) are more effective than misoprostol alone and is recommended for medical abortion at <12 weeks. The safety and effectiveness of MA drugs when provided by paramedical health workers, including nurses and ANMs have been well-established in low-resource settings [16–18] and WHO guidelines recommend nurses/ANM provision of MA at scale [19].

Off-label or over the counter sales of MA drugs at pharmacies are not permitted in Nepal [20]. There are currently six registered MA products for sale in Nepal [21]. However, studies conducted by Population Services International in Nepal (PSI/Nepal) in the year 2017 reported that there are 17 different brands of MA drugs available in the Nepal market [12]. The unregistered brands of MA drugs enter the Nepalese market from India due to the open border and are generally sold at cheaper prices than the registered brands [12].

Pharmacies in Nepal are operated by varied professionals including pharmacists, pharmacy assistants, and health workers who have received 48–72 hours orientation on pharmacy. Drug dispensing is also common among paramedics in Nepal [22]. Pharmacy workers and more informal drug sellers have been the primary sources of information and medication for the treatment and prevention of any illness in developing countries including Nepal [23]. The

Nepal Demographic and Health Survey (NDHS) 2016 data shows that 72.1% of women who had an abortion preceding 5 years adopted medical abortion procedure and about one-fifth of women who had an abortion reported receiving abortion services from pharmacies/pharmacists [24]. This is because pharmacies are more accessible for women who have limited autonomy and mobility and those in rural areas, and may offer more confidentiality than public sector health facilities [23, 25]. Women can self-administer MA drugs, complete the abortion and self-evaluate the completion of the abortion at home [26–28]. Despite these benefits, the government of Nepal does not currently recommend the pharmacy provision of MA in Nepal [13, 19].

Several studies in Nepal and elsewhere have reported that untrained pharmacy workers often provide incomplete or inaccurate information and dispense unsafe and ineffective MA drugs [29–31]. Nevertheless, women in Nepal will continue to seek MA drugs from pharmacies and since there is an increasing demand, pharmacies will continue to sell MA drugs, both registered and unregistered [32, 33]. It is therefore important to understand the pharmacy workers' practice of determining clients' gestational ages and medical eligibility for MA drugs, the accuracy of the information they provide to clients regarding the MA drug regimen, and complications including referrals in case of complications.

The Women's Health Project (WHP) is a multi-regional project implemented in countries across Africa, Asia, and Latin America whose goal is to prevent unintended pregnancies and increase access to legal abortion, post-abortion care and post-abortion contraception [34]. In support of the Government of Nepal's strategies and priorities, Population Services International (PSI) (US-based non-profit organization) has been implementing WHP since 2009 in Nepal to increase access to quality and safe MA drugs and services. PSI/Nepal has been facilitating the import and supply of Medabon® which can be used for MA [14]. PSI/Nepal also conducts medical detailing visits and provides a harm-reduction orientation to pharmacy workers who stock Medabon to educate them on the correct drug regimen and dispensing practices. This study aimed to assess the MA drug dispensing practices of private sector pharmacy workers in Nepal.

## Materials and methods

### Study design

A cross-sectional study was conducted in 267 pharmacies in Nepal where WHP was operational. As interview responses do not always reflect real-life practice, the study used a mystery client approach to collect the data [35]. In this study, mystery clients were trained people usually from the local community who visited the selected pharmacies in the assumed role of clients and then reported on their experience. Mystery clients were deployed under two scenarios in each pharmacy: i) A male mystery client seeking abortion for his wife within nine weeks of gestation and ii) A female mystery client seeking medication for an unmarried friend to induce an abortion within nine weeks of gestation.

### Study population

The study population included pharmacy workers oriented by Nepal Chemists and Druggists Association (NCDA) on harm reduction in 2017 and/or those who have received medical detailing visits from market promoters of PSI/Nepal and had a stock of Medabon [36].

### Sample size and sampling technique

The sampling frame prepared in Microsoft Excel included eligible 886 pharmacies spread across 27 districts and covering six out of seven provinces of Nepal (except Karnali province). A systematic random sampling method was used to select sample pharmacies. We calculated

the required sample size, sampling interval (n = number of pharmacies in the sampling frame/ required sample pharmacies), and a random starting point. The sample size of the required pharmacies was calculated using the formula for a cross-sectional survey. With a 95% confidence level and expected population proportion of dispensing MA drugs by pharmacy workers as 50% (conservative estimate) and a 10% margin of error, the required sample size was 268 pharmacies. After the random starting point was determined, we selected sample pharmacies after every $n^{th}$ interval. Considering a non-response rate of 5%, the final sample size was 281 pharmacies. A total of 562 mystery client visits were planned considering that each pharmacy received two mystery clients. However, only 532 mystery client visits (265 male and 267 female visits) were possible in the field. Data could not be collected from 14 pharmacies either because they were closed during the survey period despite follow-up visits or were wholesalers.

## Training and data collection

The study team recruited and trained twelve young field researchers (six males and six females) aged 20–25 years as mystery clients. They were knowledgeable in reproductive health and belonged to similar communities like the ones where the sampled pharmacies were located. This was important to assure that they would blend in and not raise suspicion, which would have otherwise jeopardized the study. All of them had some experience in survey data collection and were from a health-related academic background (bachelor's degree in public health or nursing). The mystery clients completed three days of training which included information on abortion and provision of MA drugs; survey objectives and methodology; learning and practice of predetermined scripts; and practice in the use of the digital data collection tool- SurveyCTO, to record visit reports. The mystery clients also practised role-play in pharmacies (excluded from the sample) in the Kathmandu Valley.

Mystery clients approached each sample pharmacy by saying that her/his friend/wife's menstrual period was missed for nine weeks, and probed on specifics (MA drugs, side effects, route of administration, etc.) only if the pharmacy worker did not counsel on that information. Immediately after interaction with the pharmacy worker, the mystery clients went to a secure location and fill out a standard digital survey questionnaire using the SurveyCTO application on a smartphone. The questionnaire collected information on the background characteristics of the pharmacy worker, behavior, brand, and type of MA drugs offered, counselling practices, and use of communication materials. The questionnaire also included an open-ended section for additional notes. Each interaction of the mystery client with the pharmacy workers was about 25 minutes. Confidentiality of the mystery client was maintained throughout the study to prevent any leakage of identity or information disclosed. Mystery clients visited pharmacy workers during non-peak times to reduce interference from other customers. The mystery clients paused their interaction with the pharmacy workers when other customers arrived and re-initiated it when they left. In cases where the pharmacy worker offered the purchase of the MA drug, mystery clients were instructed to buy the drug and then return it to the study team. The study team had provided the money for buying the MA drugs.

The data collection took place from September to October 2019. Once data collection was completed, a focus group discussion (FGD) was conducted with the mystery clients to explore their experiences and insights. An FGD guideline was used to explore their experience of locating pharmacies, the physical setting of the pharmacies, behavior of the pharmacy workers, quoted cost of MA drugs, suggestions from pharmacy workers for additional services and gender differences in access to MA drugs (differential behavior of pharmacy workers towards male and female mystery clients). Information collected from the FGD has been presented as verbatim to supplement quantitative data wherever appropriate.

## Data analysis

The datasets were downloaded from the SurveyCTO server and imported into Stata version 15 for analysis. Data were checked for missing values, outliers, and completeness of responses; thus, cleaned and labeled. Descriptive analysis was conducted by calculating frequency and percentages for categorical variables. Median (Interquartile range) was calculated for continuous variables that were skewed and open-ended questions were manually transcribed, coded, and analyzed under different themes and categories.

## Ethical approval

Informed consent was not obtained from the pharmacy workers during the time of data collection and neither debriefing was conducted following the data collection because the mystery clients do not reveal their identities at the time of this study. However, the memorandum of understanding (MoU) between PSI/Nepal and the pharmacies already includes their consent on mystery client survey and other data/service quality monitoring procedures. Additionally, PSI's market promoters had pre-informed pharmacy workers about the possible mystery client visits as a part of service evaluation process. No personal identifiers were disclosed anywhere in the study and strict mechanisms were put in place to protect the confidentiality of the pharmacy workers. Similarly, facility level analysis was not conducted. Ethical approval for this study including the waiver of consent procedure was obtained from the Ethical Review Board of Nepal Health Research Council (Ref: 800/2019).

## Results

### Outcome of mystery client visits

More male clients (38.1%) were dispensed MA drugs compared to female mystery clients (33.3%) Pharmacy workers refused to provide MA drugs to the mystery clients in nearly two of five (39.3%) visits (Table 1). Similarly, male pharmacy workers dispensed MA drugs (42.5% male and 35.4% female) in higher proportions than female pharmacy workers (25.7% male and 29.5% female) in mystery client visits. Interestingly, male pharmacy workers favored male while female pharmacy workers slightly preferred female while dispensing MA drugs. (Table not shown)

**Table 1. Outcome of mystery client visits.**

| Outcome of mystery client (MC) visits | Male MC visits | Female MC visits | Total MC visits |
|---|---|---|---|
| | N (%) | N (%) | N (%) |
| Dispensed and bought MA drugs[1] | 32 (12.1) | 26 (9.7) | 58 (10.9) |
| Dispensed but not bought MA drugs[2] | 69 (26.0) | 63 (23.6) | 132 (24.8) |
| Referred[3] | 37 (14.0) | 57 (21.3) | 94 (17.7) |
| Provided counseling and referred[4] | 20 (7.5) | 19 (7.1) | 39 (7.3) |
| Refused MA drugs[5] | 107 (40.4) | 102 (38.2) | 209 (39.3) |
| Total | 265 (100) | 267 (100) | 532 (100) |

[1] Was counseled, offered, and sold MA drug

[2] Was offered and counseled about MA but the client did not buy the drug

[3] Was asked to seek services elsewhere and was not counseled

[4] Was provided and offered counselling and was referred to doctor/pharmacies

[5] No counselling or, MA offered, no referrals.

Lack of evidence of prior consultation with a physician (27.5%), referral to other health facilities (21.8%), unavailability of MA drugs in the pharmacy (21.3%) and lack of prescription (16.4%) were the main reasons for refusing to dispense MA drugs to mystery clients. Likewise, unavailability of MA drugs in the pharmacy (59.4%), lack of prescription (22%) and absence of partner (12.4%) were the prime reasons provided by pharmacy workers for referring mystery clients to other clinics/pharmacies.

*"A female pharmacy worker took me into a private room within the pharmacy and asked for my signature before dispensing the MA drug. She then told me that I was too young to commit such a mistake."*- A female mystery client during her visit to Nawalparasi district

*"Some pharmacy workers said that the MA drugs were kept hidden because they were illegal. In border towns, mystery clients were also told that they could be referred to pharmacies in India."*- A male mystery client during his visit to Kailali district

## Information sought by pharmacy workers

A higher proportion of females (71.5%) were asked about the last menstrual period/months of pregnancy compared to male mystery clients (68.4%). Likewise, in 38.1% of visits, pharmacy workers inquired about whether the pregnancy had been confirmed and in 15.2% of visits, they wanted to know about how the pregnancy status was confirmed. In 2.8% of visits, the pharmacy workers asked clients about the reasons for abortion (Table 2).

## Information provided to mystery clients

During 65.1% of the visits, mystery clients were told about when to take the MA drugs, while in 66.4% of visits, they were told about the route of drug administration. Information about the possible side effects was provided in 55.9% of the visits whereas in 23.6% of visits, they were informed about complications that may occur during the abortion process. Likewise, in 13.1% of visits, they were told where to go in case of complications. (Table 3).

In 70.7% of visits, pharmacy workers provided correct information on taking MA drugs to the mystery clients (to take one 200 mg mifepristone pill on the first day orally and four 200 mg misoprostol pills after 24 hours or same time the next day either sublingually or vaginally). In most of the visits (95.7%), pharmacy workers told the mystery clients that vaginal bleeding could be heavier than during menstruation, but only half (56.5%) of them were told about stomach cramps. Vomiting and headache were the most common side effects mentioned by the pharmacy workers (Table 4).

Regarding the use of information education communication (IEC) materials, pharmacy workers displayed MA strips in one-third (33.6%) of visits while explaining the process of taking MA drugs, leaflet was shown in 17%, and leaflet was given in 6.6% of the visits while dispensing the drug (Table 5).

**Table 2. Information sought by pharmacy workers when asked for MA drugs (N = 323) (multiple choice).**

| Types of information sought from clients | Male MC visits | Female MC visits | Total MC visits |
|---|---|---|---|
|  | N (%) | N (%) | N (%) |
| Last menstrual period/month of pregnancy | 108 (68.4) | 118 (71.5) | 226 (70.0) |
| Whether the pregnancy had been confirmed | 76 (48.1) | 47 (28.5) | 123 (38.1) |
| How the pregnancy was confirmed | 22 (13.9) | 27 (16.4) | 49 (15.2) |
| Reasons for abortion | 8 (5.1) | 1(0.6) | 9 (2.8) |

**Table 3. Information provided to mystery clients about MA drugs by pharmacy workers (N = 229) (multiple choice).**

| Information provided to MCs about MA drugs* | Male MC visits | Female MC visits | Total MC visits |
|---|---|---|---|
| | N (%) | N (%) | N (%) |
| When to take the drugs | 80 (66.1) | 69 (63.9) | 149 (65.1) |
| Route of administration | 87 (71.9) | 65 (60.2) | 152 (66.4) |
| What to expect during the abortion process | 39 (32.2) | 76 (70.4) | 115 (50.2) |
| Possible side effects that may be seen during the abortion process | 67 (55.4) | 61 (56.5) | 128 (55.9) |
| Complications that may occur during the abortion process | 22 (18.2) | 32 (29.6) | 54 (23.6) |
| Where to go in case of complications | 13 (10.7) | 17 (15.7) | 30 (13.1) |
| To consult the doctor on the fifteenth day to confirm whether the abortion is complete | 7 (5.8) | 12 (11.1) | 19 (8.3) |
| The risks to the ongoing pregnancy | 6 (5) | 11 (10.2) | 17 (7.4) |

*includes those mystery clients who were dispensed and bought MA, dispensed but not bought MA drugs and those who were provided counseling and referred.

In only two out of 323 visits, the mystery clients were informed about post-abortion family planning (one implant and one intra-uterine contraceptive device) by the pharmacy workers (results not shown in table).

## Type and price of MA drugs offered to mystery clients

In nearly 80% of the visits, pharmacy workers mentioned the price of MA drugs to the mystery clients. In two out of three visits, pharmacy workers dispensed MA combi-packs, in 30.6% of visits they did not show any MA drugs, while in 3.5% of the visits they offered non-

**Table 4. Information on how to take MA drugs, and side effects.**

| Information on MA drugs | Male MC visits | Female MC visits | Total MC visits |
|---|---|---|---|
| | N (%) | N (%) | N (%) |
| **Information on how to take MA drugs* (N = 229)** | | | |
| Correct information | 86 (71.1) | 76 (70.4) | 162 (70.7) |
| Incorrect information | 35 (28.9) | 32 (29.6) | 67 (29.3) |
| **What to expect during the abortion process (N = 115)** | | | |
| Vaginal bleeding heavier than menstruation | 38 (97.4) | 72 (94.7) | 110 (95.7) |
| Stomach cramps | 13 (33.3) | 52 (68.4) | 65 (56.5) |
| Nausea | 6 (15.4) | 24 (31.6) | 30 (26.1) |
| Others** | 0 (0.0) | 2 (2.6) | 2 (1.7) |
| **Side effects mentioned (N = 128)** | | | |
| Vomiting | 24 (34.3) | 46 (65.7) | 70 (54.7) |
| Diarrhea | 5 (7.1) | 11 (15.7) | 16 (12.5) |
| Headaches | 47 (67.1) | 15 (21.4) | 62 (48.4) |
| Chills and shivering | 19 (27.1) | 15 (21.4) | 34 (26.6) |
| Transient fever | 1 (1.4) | 12 (17.1) | 13 (10.2) |
| Others | 11 (15.7) | 10 (14.3) | 21 (16.4) |

*includes those mystery clients who were dispensed and bought MA, dispensed but not bought MA drugs and those who were provided counseling and referred.

** not mentioned, the patient may faint due to heavy bleeding

**Table 5. Availability and use of IEC materials by pharmacy worker while counseling on MA (N = 229).**

| Availability and use of IEC materials | Male MC visits | Female MC visits | Total MC visits |
|---|---|---|---|
| | N (%) | N (%) | N (%) |
| Showed MA strips* while explaining the process of taking the drug | 31 (25.6) | 46 (42.6) | 77 (33.6) |
| Showed MA leaflets** while explaining how to take the drug | 16 (13.2) | 23 (21.3) | 39 (17.0) |
| Gave MA leaflets while dispensing the drug | 7 (5.8) | 8 (7.4) | 15 (6.6) |
| Showed IEC materials*** while counselling about the side effects of MA | 4 (3.3) | 4 (3.7) | 8 (3.5) |

*are the original MA drugs where pharmacy workers show the mystery clients about mifepristone and misoprostol.

**leaflets are the printed sheet of paper containing information about different aspects (side effects, indication, contra-indication, etc.) of MA drugs

*** these are IEC materials other than strips and leaflets.

combination (separately packaged misoprostol and mifepristone) drugs to the mystery clients. The median price of the MA drugs, as quoted by the pharmacy workers was NRs. 942, higher than the MRP of NRs. 800 during the time of the study (Table 6).

> *"In many cases, the quoted price was higher than the maximum retail price, and we were asked to pay more if we needed the medicine. Some asked for extra charges citing the potential risks of MA drugs."*- A male mystery client during FGD

> *"The quoted price of MA drug was usually higher for mystery clients posing as a friend of an unmarried pregnant girl. Also, the pharmacy workers said that in the absence of prescription the price of the drug will be higher."*–A female mystery client during FGD

## Discussion

Since the legalization of abortion in 2002, Nepal has made remarkable progress in the rapid scale-up of safe abortion services and is often regarded as a model for successful

**Table 6. Type and price of MA drugs.**

| Characteristics | Male MC visits | Female MC visits | Total MC visits |
|---|---|---|---|
| | N (%) | N (%) | N (%) |
| **Pharmacy workers informed clients about the price of MA drugs (N = 229)** | | | |
| Yes | 90 (74.4) | 93 (86.1) | 183 (79.9) |
| No | 31 (25.6) | 15 (13.9) | 46 (20.1) |
| **Types of MA drugs offered (N = 229)** | | | |
| Combi packs | 77 (64.2) | 73 (67.0) | 150 (65.5) |
| Separate packs | 4 (3.3) | 4 (3.7) | 8 (3.5) |
| Didn't show MA drugs | 38 (31.7) | 32 (29.4) | 70 (30.6) |
| Misoprostol Only | 1 (0.8) | 0 (0.0) | 1 (0.4) |
| **Price of MA drugs (NRs.) (N = 183)** | | | |
| 1–500 | 10 (11.0) | 15 (16.3) | 25 (13.7) |
| 501–1000 | 49 (53.8) | 42 (45.7) | 91 (49.7) |
| 1001–1500 | 23 (26.4) | 27 (29.3) | 50 (27.9) |
| >1500 | 8 (8.8) | 8 (8.7) | 15 (8.7) |
| **Median Price** | NRs. 942 | | |

implementation [37]. Recently, the Government of Nepal through the 'Right to Safe Motherhood and Reproductive Health Act 2018 [7] has further increased the indications under which abortion is legal, thus ensuring the right to safe abortion services. The National Roadmap to Improve Maternal and Newborn Health in Nepal (2019–2030) [38] acknowledges the rapidly expanding MA through pharmacies and recommends that mid-level health workers (ANMs, staff nurses and paramedics) working in the pharmacies be trained and certified to provide MA services. Among all safe abortion service users, the proportion of MA has increased over the last few years, from 53% in 2015/16 to 66% in the 2018/19 Nepali fiscal year [39]. Pharmacies are thus increasingly becoming the desired pathway for accessing MA drugs for abortion services among women in Nepal.

The over-the-counter sale of MA drugs is prohibited under the existing regulations of the Government of Nepal. Regardless, pharmacy workers dispensed MA drugs to mystery clients in more than one-third of the visits. In contrast, a study conducted in Kenya reported that only 4.3% of pharmacy workers provided abortion methods [40]. Nearly one in five mystery clients in our study were refused MA drug by the pharmacy workers during their visits because they did not have a prescription. Zambia in 2011 reported a higher proportion of refusal where nearly two in five mystery clients were not provided with the MA drugs because of legal reasons of not having the prescription [41].

In our study, male mystery clients were offered MA drugs more easily without much probing compared to females in the same pharmacies which reflects gender discrepancies in behavior of pharmacy workers. This might be due to unequal power relations between males and females in Nepal's patriarchal dominant society and abortion stigma which also affects women's reproductive health decision making [33, 42, 43]. A good interpersonal relationship between a patient and provider—as characterized by mutual respect, openness and a balance in their respective roles in decision-making–is an important marker of quality of care [44]. However, gender bias as such could affect trust in the health facilities and might force women to access unsafe abortions. Importantly, pharmacy workers should be trained to counsel male clients so that correct information be provided to MA user [45].

Given the effectiveness of MA drugs up to only nine to 10 weeks of gestation, the time since last menstruation must be investigated to determine the gestational age of the fetus. Ideally, confirmation of pregnancy by gynecological examination, ultrasound scan or biological tests is recommended. In most visits, pharmacy workers asked about the last menstrual period/month of pregnancy of their friends/wives to determine the gestational age. This result is similar to that of a study conducted in Uttar Pradesh, India where 76.6% of mystery clients were asked about their last menstrual period to confirm their gestational age [46].

Information on the correct time for taking MA drugs, route of administration and dosage, abortion process, side effects/complications and appropriate referral information are fundamental for ensuring safe MA. Studies elsewhere have shown that a significant proportion of pharmacy workers fail to provide this information to the clients while dispensing MA drugs [30, 32, 40, 46–50]. One possible reason might be that the pharmacy workers might not be adequately trained in communicating this vital information while dispensing MA drugs [45]. In seven out of ten visits, pharmacy workers in this study correctly informed the mystery clients about the timing, dose and route of administration of MA drugs compared to a very limited proportion of the pharmacy workers in India (35.3%), Kenya (6.3%) and Zambia (21%) [40, 41, 50]. The harm reduction orientation provided by NCDA and medical detailing visits by PSI/Nepal's staff does cover these topics which might have improved pharmacy workers' skills in providing information about MA.

Pharmacy workers dispensed MA drugs to more than a quarter of mystery clients during their visits in this study. Most of them were offered the combi-pack (mifepristone and

misoprostol) drugs for MA and more than half of them were informed about the side effects of MA drugs. But still, some of the pharmacies are offering separate packs for medical abortion which are considered illegal in Nepal. This might be because of the high influx of those medical drugs from India and high-cost margins [46]. The quality and source of these drugs are also questionable which can affect women's health. We can assume that clients coming to buy MA drugs from pharmacies are usually seeking clandestine and quick abortion solutions at any cost. Pharmacy workers can take advantage of their desperate situation by quoting a price that is higher than the maximum retail price (MRP) and someone in dire need can be expected to oblige. In this study, the median price of MA drugs (NRs. 942) quoted by the pharmacy workers was much higher than the MRP (NRs. 800 similar to a study done in Pokhara, Nepal where seven out of nine MA drugs were sold at higher price than the labelled [32]. According to the mystery clients, pharmacy workers quoted a higher price citing the potential risks with MA drugs and the lack of prescription. The Department of Drug Administration (DDA) under the MoHP thus need to regulate the registration and pricing of drugs in Nepal.

By using a mystery client approach, this study has revealed the MA drug dispensing practices of pharmacy workers in Nepal and has some important implications. Even though pharmacy provision of MA drugs is not recommended in Nepal [7], pharmacy workers dispensed MA drugs and provided correct information most of the time in this study. Most of these pharmacies stock Medabon, a combination drug and have received training from NCDA and PSI/Nepal on the proper use and dispensing of MA drugs. This indicates that with proper training and orientation pharmacy workers may be able to safely provide MA drugs over the counter without a prescription. Similar recommendation was provided by a study done in Eastern Nepal in 2011 where pharmacy workers after harm reduction training provided correct information on use of medical abortion drug [51]. In a previous study done in Nepal, MA services provided by pharmacy workers were acceptable to women and were satisfied with the service they had received [20]. However, it will be essential to regularly monitor the pharmacies to check if they are following the national guidelines and are engaging in lawful practices.

This study has some limitations. First, there was a possibility of misreporting or recall bias by the mystery clients, although it was minimized through the completion of the structured survey questionnaire immediately after their exit from the pharmacy. Second, the identity of the mystery clients could have been compromised and the pharmacy workers may have behaved differently toward them. We minimized this by recruiting mystery clients from the local community and ensuring the local dialect and dress-up during their visit to the pharmacy. Third, the sample of pharmacies assessed in this study may not be representative of all pharmacies in Nepal. Since the pharmacy workers were trained by NCDA and PSI/Nepal in the past, their MA drug dispensing behavior can be assumed to be better than the rest of the pharmacies in Nepal.

The ubiquitous presence of pharmacies and relatively easier access in hard-to-reach areas, make pharmacy provision of MA drugs a viable option for bringing MA services closer to women in need. Recently in 2022, WHO in its new abortion care guideline recommended pharmacy provision of MA including self-management for induced abortion up to 10 weeks which was previously not recommended [52]. The new guideline recognized the skills and knowledge of these cadres for dispensing MA drugs. Given the involvement of varied professionals in dispensing drugs in Nepal at the pharmacy level, it would be important to orient these cadres especially with non-pharmacy background in dispensing MA drugs. During the COVID-19 pandemic, the family welfare division under the MoHP through the 'Interim Guidance for RMNCH services in COVID-19 Pandemic' [53] allowed all registered pharmacists to store and dispense MA drugs on prescription. Previously, only pharmacists within 100 meters of a registered safe abortion site were allowed to do so [13]. Although interim, this was a

progressive policy change that if replicated in regular guidelines would further improve abortion access.

Considering the existing recommendations provided by WHO and country-level evidence, Government of Nepal should reflect and review the existing policies related to medical management of induced abortion to ensure universal access to sexual and reproductive health services. The relatively confidential setup and ease of access within communities make pharmacies an attractive option for women who want to seek abortion services. It would be best not to keep these services in the grey area as far as legislation is concerned but regulate them through adequate training and supervision.

## Conclusions

In more than one-third of the visits, pharmacy workers dispensed MA drugs to the mystery clients regardless of the legal provision for dispensing MA drugs upon prescription. In seven out of ten visits, pharmacy workers correctly informed the mystery clients about the timing, dose, and route of administration of MA drugs. The figures could be different in pharmacies who have not received harm reduction training or medical detailing visits. Engaging pharmacy workers as first contacts of women for safe abortion services, particularly MA would thus be an effective strategy to reduce the incidence of unsafe abortion. For this, regular and effective training/orientation of pharmacy workers is pivotal which will also ensure conformity to existing legal standards. Training/orientation program should be focused on critical areas, including the provision of basic information on MA drugs, complications, and referral in case of complications, screening to determine a client's eligibility for MA and informing clients about the possible side effects, mode of action, possible danger signs and the options for post-abortion family planning.

## Acknowledgments

The authors would like to acknowledge the study participants for their participation in the survey. Similarly, we would like to thank field research team, and Health Foundation Nepal for their support in the implementation of the survey.

## Author Contributions

**Conceptualization:** Anil Sigdel, Mirak Raj Angdembe, Nilaramba Adhikari, Alina Maharjan, Mahesh Paudel.

**Data curation:** Anil Sigdel.

**Formal analysis:** Anil Sigdel, Pratik Khanal.

**Funding acquisition:** Anil Sigdel, Mirak Raj Angdembe, Alina Maharjan.

**Investigation:** Anil Sigdel, Mirak Raj Angdembe.

**Methodology:** Anil Sigdel, Mirak Raj Angdembe, Nilaramba Adhikari, Alina Maharjan.

**Project administration:** Anil Sigdel, Mirak Raj Angdembe, Alina Maharjan.

**Resources:** Anil Sigdel, Mirak Raj Angdembe.

**Software:** Anil Sigdel.

**Supervision:** Anil Sigdel, Mirak Raj Angdembe, Alina Maharjan, Mahesh Paudel.

**Validation:** Anil Sigdel, Mirak Raj Angdembe.

**Writing – original draft:** Anil Sigdel, Pratik Khanal.

**Writing – review & editing:** Mirak Raj Angdembe, Nilaramba Adhikari, Alina Maharjan, Mahesh Paudel.

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
