## [Decision Letter · Decision Letter 0]

8 Aug 2022

PONE-D-22-11126Medical abortion drug dispensing practices among private pharmacy workers in Nepal: a mystery client studyPLOS ONE

Dear Dr. Pratik Khanal,

Thank you for submitting your manuscript to PLOS ONE. After careful consideration, we feel that it has merit but does not fully meet PLOS ONE’s publication criteria as it currently stands. Therefore, we invite you to submit a revised version of the manuscript that addresses the points raised during the review process. The revised version of the manuscript must addresses all points raised during the review process by #Reviewer1 and #Reviewer2, besides the following points:

Introduction

Please review and clarify the following sentences:

-WHO Model List of Essential Medicine (reference 15) does not present data about the effectiveness of MA drugs; it just list the drugs.

-“The Nepal Demographic and Health Survey (NDHS) 2016 data shows that about one fifth of women who had an abortion reported receiving abortion services from pharmacies”. According to the NDHS, from pharmacies and health providers, right?

Methods

“A total of 15 pharmacies were considered as non-response as these pharmacies were closed even after approaching three times during mystery client visits.” Did not the information on opening hours of pharmacies available to mystery clients before the visit?

Discussion and Conclusion

“Recently in 2022, WHO in its new abortion care guideline recommended for pharmacy provision of MA including self-management.” It is essential to present details of this recommendation according to the WHO guideline (gestational condition, medical abortion regimens, etc.)

You pointed out in the Discussion that the study population selected (“pharmacy workers oriented by NCDA on harm reduction in 2017 and/or those who have received medical detailing visits from market promoters of PSI/Nepal”) would not represent the Nepal pharmacies.  The selection bias needs to be warned in the abstract and conclusion, mentioning that this selective sample does not represent all pharmacies in Nepal.

Tables: Consider joint tables 4, 5, and 6.

We look forward to receiving your revised manuscript.

Kind regards,

Tatiane da Silva Dal Pizzol, Ph.D.

Academic Editor

PLOS ONE

Journal Requirements:

2. Thank you for indicating that informed consent was not obtained due to the mystery client aspect of the study, and your statement:  Ethical approval for this study including the permission for consent procedure was obtained from the Ethical Review Board of Nepal Health Research Council (Ref: 800/2019)" Please could you revise this statement to indicate whether the IRB had waived the need for informed consent.

3. Thank you for stating in your financial disclosure:  

The study is funded under the Women's Health Project implemented by Population Service International. The project receives fund from anonymous donors. The funders had no role in study design, data collection and analysis, decision to publish, or preparation of the manuscript. 

PLOS ONE requires you to include in your manuscript further information about the funder so that any relevant competing interests can be assessed. Please respond to the following questions:

1. Please state whether any of the research costs or authors' salaries were funded, in whole or in part, by a tobacco company (our policy on tobacco funding is at http://journals.plos.org/plosone/s/disclosure-of-funding-sources)  

2. Please state whether the donor has any competing interests in relation to this work (see http://journals.plos.org/plosone/s/competing-interests) . 

3. Please state whether the identity of the donor might be considered relevant to editors or reviewers’ assessment of the validity of the work.

4. If the donors have no perceived or actual competing interests, please state: “The authors are not aware of any competing interests”. 

This information should be included in your cover letter. We will amend your financial disclosure and competing interests on your behalf.

Reviewers' comments:

Reviewer's Responses to Questions

**Comments to the Author**

1. Is the manuscript technically sound, and do the data support the conclusions?

Reviewer #1: Yes

Reviewer #2: Yes

2. Has the statistical analysis been performed appropriately and rigorously? 

Reviewer #1: Yes

Reviewer #2: N/A

3. Have the authors made all data underlying the findings in their manuscript fully available?

Reviewer #1: Yes

Reviewer #2: Yes

4. Is the manuscript presented in an intelligible fashion and written in standard English?

Reviewer #1: Yes

Reviewer #2: Yes

5. Review Comments to the Author

Reviewer #1: The authors report on an important public health issue for women globally and especially in low- and middle-income countries. The role of pharmacies and their staff providing an often-illegal medication for abortion and the consequences of their lack of quality care for maternal mortality and morbidity are vital issues for critical research. There is however a lack of clarity in some descriptions of the methodology and additional suggestions for improvement below.

Introduction

This intro does give a good overview of the legal MA situation in Nepal and makes a good argument for the current study; however I find it odd that you haven’t referenced relevant studies in Nepal and India published in this journal either here or in the discussion – e.g.

1. Medical abortion kit dispensing practices of community pharmacies in Pokhara Metropolitan, Nepal Nim Bahadur Dangi, Sangam Subedi, Mahasagar Gyawali, Aashish Bhattarai, Tulsi Ram Bhandari | published 13 Jan 2021 PLOS ONE

2. Delivering Medical Abortion at Scale: A Study of the Retail Market for Medical Abortion in Madhya Pradesh, India Powell-Jackson T, Acharya R, Filippi V, Ronsmans C (2015) Delivering Medical Abortion at Scale: A Study of the Retail Market for Medical Abortion in Madhya Pradesh, India. PLOS ONE 10(3):0120637. https://doi.org/10.1371/journal.pone.0120637

3. Acceptability of Home-Assessment Post Medical Abortion and Medical Abortion in a Low-Resource Setting in Rajasthan, India. Secondary Outcome Analysis of a Non-Inferiority Randomized Controlled Trial Mandira Paul, Kirti Iyengar, Birgitta Essén, Kristina Gemzell-Danielsson, Sharad D. Iyengar, Johan Bring, Sunita Soni, Marie Klingberg-Allvin Research Article | published 01 Sep 2015 PLOS ONE https://doi.org/10.1371/journal.pone.0133354

4. Pathways to seeking medication abortion care: A qualitative research in Uttar Pradesh, India

Aradhana Srivastava, Malvika Saxena, Joanna Percher, Nadia Diamond-Smith Research Article | published 13 May 2019 PLOS ONE

Materials and methods

Both the population and mystery shopper methodology seem highly appropriate for this study.

Study design

1. However, if the study population is ‘oriented’ (please clarify what that means) by NCDA and have received detailing from PSI and stock Medabon, what is the relation between the 266 pharmacies surveyed (mentioned in the opening line 116 on P 6) and the 886 pharmacies outlined as the eligible sampling frame on p7? Please give the year and numbers of pharmacies where PSI did medical detailing.

2. Page 7. L127. What is the relevance of this reference (33) to the sentence? Suggest drop

Sampling size and sampling technique –

3. Please reference your random sampling method. How was it implemented?

4. L.133 what variable was your 50% estimate based on – can you please state? For example, was it medication dispensed, gestational age asked?

Training and data collection. The training you have offered your mystery shopper (MS) staff sounds thorough. Your focus group discussion after was also a good strategy to gather feedback data. Can you please however add more clarity to the following:

5. Did you collect gender of pharmacy worker and breakdown the responses further? It would be relevant to see whether there were gendered behaviours and attitudes between differently gendered MS and pharmacy workers for a stigmatised, gendered and illegal activity.

6. What pharmacy worker behaviours did you record – can you please clarify even if giving examples in brackets. Did it include attitudes?

7. It appears that ‘counselling practices’ means giving vital information? Pls give examples and clarify what you mean.

8. Also what factors are included in ‘quality of care’?

9. How did you maintain confidentiality? Do you mean the MS never gave any personal information?

Data analysis and ethics are all appropriate.

10. Line 188 Can you please include what timeframe PSI gave pharmacy-workers about the possible mystery client (MC) visits (i.e. how long before you conducted the survey?)

Results

11. As indicated above, it would be helpful if you described, even if not tabulated whether there was a difference in female to female MC provision or female to male MC provision etc. Especially as more women were referred as well.

12. The information about gender of pharmacy worker would seem important if asking about LMP also. Consider providing this information if possible.

13. You provide a quote about cesarian births and MA (P.14 L242), but there does not appear to be a reason for it, not comment about whether this was wise or unwise information.

14. As you are commenting on price (p.17), it is useful to provide the either mean retail of legal provision or range in US dollars and Nepalese currency to compare to the median MCs are asked for. You could do this in the intro or on Table 8 or below it, as well as in the discussion.

Discussion

15. P19 L 302, suggest ‘provocations’ is unclear. Can you clarify whether you mean judgmental questions or comments?

16. You make very important comments here about the gender disparities. Can you comment on how this was enacted with genders between pharmacy workers and MCs as previously suggested?

17. P.19. Suggest moving para re over-the-counter sales L310 above L302 re gender of provider, so you introduce general findings before gender specific provision.

18. P. 19 L 314. The sentence reads more clearly if you omit ‘to provide’.

19. P. 19 L 316-7. Suggest ‘..not provided with the MA drugs because they did not have a prescription’.

20. P 20 L 333-5 Please detail what proportion of these pharmacies were provide with training and when and then comment on the role of training.

21. P 21 L343 MRP – please tell us what this is so we can compare.

22. Suggest you comment that if drugs are illegal, their quality and sources are also suspect and that this is dangerous, as well as price gouging possible

23. P22, L 373, no need for ‘for’ pharmacy provision. Omit ‘for’.

This is an important study. Well done

Reviewer #2: This study aimed to assess the medical abortion

drug dispensing practices of private sector pharmacy workers in Nepal using the mystery client method.

Introduction

Lines 94-98: The study involves pharmacy workers, but the authors cite a reference with data obtained from pharmacists (Rogers et al. 2019 - reference 30).

A. Please explain in the article how the service provided in Nepalese pharmacies is carried out (only by pharmacists, only by attendants?)

B. I suggest lines 94-98 revision: it was not clear whether what is being highlighted is the different performance between these two professionals or between the different scenarios (pharmacy-medication x surgical procedure).

Materials and Methods

Lines 137-138: I am unsure if this sentence is clear. I suggest text review.

Results

This section presents the same results in table and text. Please, review.

Discussion and Conclusion

The pharmacist's role in pharmacies should also be discussed. The introduction does not give us elements for a complete comprehension of the performance of this professional in private pharmacies in Nepal.

Please, present and discuss the limitations of the research.

6. PLOS authors have the option to publish the peer review history of their article (what does this mean?). If published, this will include your full peer review and any attached files.

Reviewer #1: **Yes: **Angela Taft MPH PhD

Reviewer #2: **Yes: **Elisangela Costa Lima

---

## [Author Response · Author response to Decision Letter 0]

21 Oct 2022

October 04, 2022

To,

Editorial Board

Plos One

Subject: Submission of revised manuscript PONE-D-22-11126 titled ‘Medical abortion drug dispensing practices among private pharmacy workers in Nepal: a mystery client study’

The study team would like to thank editor and reviewers for providing their feedback on the manuscript and we are confident that the reviewer’s comment has helped to improve the quality of the manuscript. Please find the response to the reviewer’s comment below. 

Editorial comments

Introduction

Please review and clarify the following sentences:

WHO Model List of Essential Medicine (reference 15) does not present data about the effectiveness of MA drugs; it just list the drugs.

Thank you for the observation. We have removed the reference and added another reference related to the effectiveness of medical abortion drugs. 

“The Nepal Demographic and Health Survey (NDHS) 2016 data shows that about one fifth of women who had an abortion reported receiving abortion services from pharmacies”. According to the NDHS, from pharmacies and health providers, right?

Thank you for the query. As per NDHS, 19% received abortion services from pharmacies/pharmacists while 71% received services from doctors and nurses, and the remaining received from others. 

Methods

“A total of 15 pharmacies were considered as non-response as these pharmacies were closed even after approaching three times during mystery client visits.” Did not the information on opening hours of pharmacies available to mystery clients before the visit?

The opening hours of pharmacies were not available to mystery clients before the visit. The non-response was because these pharmacies were closed during the study period or were wholesalers. 

Discussion and Conclusion

“Recently in 2022, WHO in its new abortion care guideline recommended for pharmacy provision of MA including self-management.” It is essential to present details of this recommendation according to the WHO guideline (gestational condition, medical abortion regimens, etc.)

Thank you for the suggestion. We have now added this information.

You pointed out in the Discussion that the study population selected (“pharmacy workers oriented by NCDA on harm reduction in 2017 and/or those who have received medical detailing visits from market promoters of PSI/Nepal”) would not represent the Nepal pharmacies. The selection bias needs to be warned in the abstract and conclusion, mentioning that this selective sample does not represent all pharmacies in Nepal.

Thank you for the suggestion. We now have added the information in the abstract and conclusion section. 

Tables: Consider joint tables 4, 5, and 6.

Thank you. We have joined table 4,5 and 6. 

Journal Requirements:

We have ensured that the manuscript meets PLOS ONE’s style requirements. 

2. Thank you for indicating that informed consent was not obtained due to the mystery client aspect of the study, and your statement: Ethical approval for this study including the permission for consent procedure was obtained from the Ethical Review Board of Nepal Health Research Council (Ref: 800/2019)" Please could you revise this statement to indicate whether the IRB had waived the need for informed consent.

We have revised the ethics statement to indicate that the Nepal’s ethical review board had waived the need for informed consent. 

3. Thank you for stating in your financial disclosure: 

The study is funded under the Women's Health Project implemented by Population Service International. The project receives fund from anonymous donors. The funders had no role in study design, data collection and analysis, decision to publish, or preparation of the manuscript. 

PLOS ONE requires you to include in your manuscript further information about the funder so that any relevant competing interests can be assessed. Please respond to the following questions:

1. Please state whether any of the research costs or authors' salaries were funded, in whole or in part, by a tobacco company (our policy on tobacco funding is at http://journals.plos.org/plosone/s/disclosure-of-funding-sources) 

Neither the study nor the author’s salaries are funded in whole or in part by a tobacco company. All the authors are PSI Nepal staffs. 

2. Please state whether the donor has any competing interests in relation to this work (see http://journals.plos.org/plosone/s/competing-interests) . 

The authors declare no competing interests in relation to this work. 

3. Please state whether the identity of the donor might be considered relevant to editors or reviewers’ assessment of the validity of the work.

The donors are anonymous and are related to philanthropic organizations. 

4. If the donors have no perceived or actual competing interests, please state: “The authors are not aware of any competing interests”. 

The authors are not aware of any competing interests. 

This information should be included in your cover letter. We will amend your financial disclosure and competing interests on your behalf.

Thank you. We will include this information in our cover letter. 

The corresponding author’s ORCID iD has been indicated in the Editorial Manager. 

5. Review Comments to the Author

Reviewer #1: The authors report on an important public health issue for women globally and especially in low- and middle-income countries. The role of pharmacies and their staff providing an often-illegal medication for abortion and the consequences of their lack of quality care for maternal mortality and morbidity are vital issues for critical research. There is however a lack of clarity in some descriptions of the methodology and additional suggestions for improvement below.

Introduction

This intro does give a good overview of the legal MA situation in Nepal and makes a good argument for the current study; however I find it odd that you haven’t referenced relevant studies in Nepal and India published in this journal either here or in the discussion – e.g.

1. Medical abortion kit dispensing practices of community pharmacies in Pokhara Metropolitan, Nepal Nim Bahadur Dangi, Sangam Subedi, Mahasagar Gyawali, Aashish Bhattarai, Tulsi Ram Bhandari | published 13 Jan 2021 PLOS ONE

2. Delivering Medical Abortion at Scale: A Study of the Retail Market for Medical Abortion in Madhya Pradesh, India Powell-Jackson T, Acharya R, Filippi V, Ronsmans C (2015) Delivering Medical Abortion at Scale: A Study of the Retail Market for Medical Abortion in Madhya Pradesh, India. PLOS ONE 10(3):0120637. https://doi.org/10.1371/journal.pone.0120637

3. Acceptability of Home-Assessment Post Medical Abortion and Medical Abortion in a Low-Resource Setting in Rajasthan, India. Secondary Outcome Analysis of a Non-Inferiority Randomized Controlled Trial Mandira Paul, Kirti Iyengar, Birgitta Essén, Kristina Gemzell-Danielsson, Sharad D. Iyengar, Johan Bring, Sunita Soni, Marie Klingberg-Allvin Research Article | published 01 Sep 2015 PLOS ONE https://doi.org/10.1371/journal.pone.0133354

4. Pathways to seeking medication abortion care: A qualitative research in Uttar Pradesh, India

Aradhana Srivastava, Malvika Saxena, Joanna Percher, Nadia Diamond-Smith Research Article | published 13 May 2019 PLOS ONE

Thank you very much for the suggestion. We have missed to include these useful references which now have been included in the paper. 

Materials and methods

Both the population and mystery shopper methodology seem highly appropriate for this study.

Study design

1. However, if the study population is ‘oriented’ (please clarify what that means) by NCDA and have received detailing from PSI and stock Medabon, what is the relation between the 266 pharmacies surveyed (mentioned in the opening line 116 on P 6) and the 886 pharmacies outlined as the eligible sampling frame on p7? Please give the year and numbers of pharmacies where PSI did medical detailing.

The sampling frame included all pharmacies which had a stock of Medabon. The pharmacies surveyed are sampled pharmacies from the sampling frame. All these pharmacies have received medical detailing visits at least once a year since they had stocked Medabon. NCDA had provided harm reduction training in 2017 to some of these pharmacies. 

2. Page 7. L127. What is the relevance of this reference (33) to the sentence? Suggest drop

We have decided to include this reference as this provides reference for the readers to understand more about the mystery client approach. 

Sampling size and sampling technique –

3. Please reference your random sampling method. How was it implemented?

Thank you for the feedback. We have included a detailed information about the sampling strategy in the sample size section. 

4. L.133 what variable was your 50% estimate based on – can you please state? For example, was it medication dispensed, gestational age asked?

The 50% estimate was based on medication dispensing behavior of pharmacy workers. Since we did not have recent estimate, we used the conservative estimate of 50% to obtain maximum sample of pharmacies. 

Training and data collection. The training you have offered your mystery shopper (MS) staff sounds thorough. Your focus group discussion after was also a good strategy to gather feedback data. Can you please however add more clarity to the following:

5. Did you collect gender of pharmacy worker and breakdown the responses further? It would be relevant to see whether there were gendered behaviours and attitudes between differently gendered MS and pharmacy workers for a stigmatised, gendered and illegal activity.

Yes, we collected gender of pharmacy workers in the study. In the 267 pharmacies surveyed, 69% were male and 31% were female pharmacy workers. During the further analysis, we found that male pharmacy workers dispensed MA drugs in higher proportions to the mystery clients than the female pharmacy workers. Interestingly, male pharmacy workers favored male while female pharmacy workers slightly preferred female while dispensing MA drugs. We have added this information in the manuscript. 

6. What pharmacy worker behaviours did you record – can you please clarify even if giving examples in brackets. Did it include attitudes?

Dispensing behavior of pharmacy workers towards medical abortion seeking clients were recorded through questionnaire. Focus group discussion collected counseling related behavior of pharmacy workers towards clients which was immediately recorded after the interaction between provider and client. 

7. It appears that ‘counselling practices’ means giving vital information? Pls give examples and clarify what you mean.

Counseling practices are related to the dosage and administration of MA drugs, what to expect during the abortion process, possible side effects and complications, where to go in case of complications, what to do to assess completion of abortion, and post-abortion family planning use. 

8. Also what factors are included in ‘quality of care’?

We assessed the quality of care received by the mystery clients on different domains of services received through Likert scale. These were related to attitude of pharmacy workers, attitude of other staff, satisfaction with the counseling, had the opportunity to talk with the provider, had the opportunity to tell what they wanted, had the opportunity to ask questions and comfort level when talking with the provider. Similarly, perceived quality of physical domain of pharmacy such as visual and auditory privacy, adequate privacy at reception, waiting time, comfortable waiting area, location of the facility, and overall satisfaction with the facility was also assessed. However, we have not included this information on the manuscript. Hence, we would like to remove this variable. 

9. How did you maintain confidentiality? Do you mean the MS never gave any personal information?

Yes, mystery client never revealed their identity to the pharmacy worker during their interaction. 

Data analysis and ethics are all appropriate.

Thank you for your observation. 

10. Line 188 Can you please include what timeframe PSI gave pharmacy-workers about the possible mystery client (MC) visits (i.e. how long before you conducted the survey?)

It was during the facility contracting process that PSI gave information to the pharmacy workers about the possible mystery client visits. Facilities are usually contracted at the starting of the year (January). For facilities which are not a part of the PSI network but stock medabon, market promoters conduct medical detailing visits at least once a year (usually in the first half of the year, January-June) who also inform about the mystery client visits as a part of quality evaluation procedure. Data collection of the study was done in between September to October 2019. 

Results

11. As indicated above, it would be helpful if you described, even if not tabulated whether there was a difference in female to female MC provision or female to male MC provision etc. Especially as more women were referred as well.

Thank you for the suggestion. We have included this information in the dispensing behavior of pharmacy workers. 

12. The information about gender of pharmacy worker would seem important if asking about LMP also. Consider providing this information if possible.

There was not much difference in asking about LMP by gender of pharmacy workers. While male pharmacy workers asked about LMP in 71% of the visits while female pharmacy workers inquired about LMP in 68% of the visits. 

13. You provide a quote about cesarian births and MA (P.14 L242), but there does not appear to be a reason for it, not comment about whether this was wise or unwise information.

We have decided to not to use this quote considering ambiguity. 

14. As you are commenting on price (p.17), it is useful to provide the either mean retail of legal provision or range in US dollars and Nepalese currency to compare to the median MCs are asked for. You could do this in the intro or on Table 8 or below it, as well as in the discussion.

Thank you for the feedback. We have included in the results section. 

Discussion

15. P19 L 302, suggest ‘provocations’ is unclear. Can you clarify whether you mean judgmental questions or comments?

We have rephrased the word ‘provocations’ to ‘probing’. This indicates towards the dispensing behavior of pharmacy workers. 

16. You make very important comments here about the gender disparities. Can you comment on how this was enacted with genders between pharmacy workers and MCs as previously suggested?

We have now included information about variation in dispensing behavior towards male and female mystery clients by gender of pharmacy workers. We found that male pharmacy workers dispense MA drugs to the mystery clients in higher proportions than the female pharmacy workers. Considering gender of mystery clients, female pharmacy workers dispensed MA drug slightly higher to the female mystery clients than the male mystery clients. 

17. P.19. Suggest moving para re over-the-counter sales L310 above L302 re gender of provider, so you introduce general findings before gender specific provision.

Done

18. P. 19 L 314. The sentence reads more clearly if you omit ‘to provide’.

Done

19. P. 19 L 316-7. Suggest ‘..not provided with the MA drugs because they did not have a prescription’.

Done

20. P 20 L 333-5 Please detail what proportion of these pharmacies were provide with training and when and then comment on the role of training.

All these pharmacies received medical detailing visits from market promoters since they had a stock of Medabon- a MA drug. Among the pharmacies surveyed, 39 (14.6%) pharmacies were oriented on harm reduction orientation. The role of both these activities is to educate pharmacy workers about the dosage and administration of MA drugs along with its side effects so that they can counsel clients while dispensing. 

21. P 21 L343 MRP – please tell us what this is so we can compare.

The information about the MRP has now been included. The MRP was NRs. 800 during the time of the survey while the median price offered to the clients was NRs. 942. 

22. Suggest you comment that if drugs are illegal, their quality and sources are also suspect and that this is dangerous, as well as price gouging possible

We appreciate your thoughts and have added this statement. The quality and source of these drugs are also questionable which can affect women’s health.

23. P22, L 373, no need for ‘for’ pharmacy provision. Omit ‘for’.

Done

This is an important study. Well done

Thank you for your acknowledgement on our paper. 

Reviewer #2: This study aimed to assess the medical abortion drug dispensing practices of private sector pharmacy workers in Nepal using the mystery client method.

Introduction

Lines 94-98: The study involves pharmacy workers, but the authors cite a reference with data obtained from pharmacists (Rogers et al. 2019 - reference 30).

We have removed this reference. 

A. Please explain in the article how the service provided in Nepalese pharmacies is carried out (only by pharmacists, only by attendants?)

Thank you for the feedback, We have included this information in introduction section. 

Pharmacies in Nepal are operated by varied professionals including pharmacists, pharmacy assistants, and health workers who have received 48-72 hours orientation on pharmacy. Drug dispensing is also common among paramedics in Nepal

B. I suggest lines 94-98 revision: it was not clear whether what is being highlighted is the different performance between these two professionals or between the different scenarios (pharmacy-medication x surgical procedure).

We have removed the statement and hence the reference. 

Materials and Methods

Lines 137-138: I am unsure if this sentence is clear. I suggest text review.

We have revised the sentence and we hope that the sentence is now clear. 

Results

This section presents the same results in table and text. Please, review.

The results section is now slightly revised. As the table is self-explanatory, we have not described the results to a full extent. 

Discussion and Conclusion

The pharmacist's role in pharmacies should also be discussed. The introduction does not give us elements for a complete comprehension of the performance of this professional in private pharmacies in Nepal.

Thank you for the suggestion. Pharmacies in Nepal are operated by varied professionals ranging from pharmacists (those with bachelor’s degree in pharmacy), pharmacy assistants (3 years course after school level), other health workers with 48-72 hours of orientation on dispensing, and even paramedics without any orientation. Pharmacists are mostly available in hospital pharmacies. However, where pharmacies are operated standalone, pharmacists are almost non-existent. Given the involvement of varied professionals in dispensing drugs in Nepal at the pharmacy level, it would be important to orient these cadres especially with non-pharmacy background in dispensing MA drugs. We now have included this information in the introduction and discussion section and hence it would allow readers to obtain some insights on how pharmacies are operated in Nepal. 

Please, present and discuss the limitations of the research.

Thank you for the suggestion. We have discussed the limitations of the study in the discussion section.

---

## [Decision Letter · Decision Letter 1]

10 Nov 2022

Medical abortion drug dispensing practices among private pharmacy workers in Nepal: a mystery client study

PONE-D-22-11126R1

Dear Dr. Pratik Khanal,

We’re pleased to inform you that your manuscript has been judged scientifically suitable for publication and will be formally accepted for publication once it meets all outstanding technical requirements.

Kind regards,

Tatiane da Silva Dal Pizzol, Ph.D.

Academic Editor

PLOS ONE

Additional Editor Comments (optional):

Reviewers' comments:

Reviewer's Responses to Questions

**Comments to the Author**

1. If the authors have adequately addressed your comments raised in a previous round of review and you feel that this manuscript is now acceptable for publication, you may indicate that here to bypass the “Comments to the Author” section, enter your conflict of interest statement in the “Confidential to Editor” section, and submit your "Accept" recommendation.

Reviewer #1: All comments have been addressed

2. Is the manuscript technically sound, and do the data support the conclusions?

Reviewer #1: Yes

3. Has the statistical analysis been performed appropriately and rigorously? 

Reviewer #1: Yes

4. Have the authors made all data underlying the findings in their manuscript fully available?

Reviewer #1: Yes

5. Is the manuscript presented in an intelligible fashion and written in standard English?

Reviewer #1: Yes

6. Review Comments to the Author

Reviewer #1: I did find it difficult to see where comments have been changed without any highlighting or track changes.

Nevertheless, I believe this reads better and more clearly. I am sorry that I did not request this in the previous review but I do believe it would help if you could indicate what proportion and number of clinics were urban, rural and remote, as this helps to strengthen your comment about access.

7. PLOS authors have the option to publish the peer review history of their article (what does this mean?). If published, this will include your full peer review and any attached files.

Reviewer #1: **Yes: **Angela Taft MPG PhD Adjunct Professor

---

## [Editor Report · Acceptance letter]

14 Nov 2022

PONE-D-22-11126R1 

Medical abortion drug dispensing practices among private pharmacy workers in Nepal: a mystery client study 

Dear Dr. Khanal:

I'm pleased to inform you that your manuscript has been deemed suitable for publication in PLOS ONE. Congratulations! Your manuscript is now with our production department. 

Kind regards, 

on behalf of

Dr. Tatiane da Silva Dal Pizzol 

Academic Editor

PLOS ONE